# Construction of Brain Metastasis Prediction Model and Optimization of Prophylactic Cranial Irradiation Selection for Limited-Stage Small-Cell Lung Cancer

**DOI:** 10.3390/cancers14194906

**Published:** 2022-10-07

**Authors:** Qing Hou, Bochen Sun, Ningning Yao, Yu Liang, Xin Cao, Lijuan Wei, Jianzhong Cao

**Affiliations:** Department of Radiotherapy, Shanxi Province Cancer Hospital/Shanxi Hospital Affiliated to Cancer Hospital, Chinese Academy of Medical Sciences/Cancer Hospital Affiliated to Shanxi Medical University, Taiyuan 030013, China

**Keywords:** small-cell lung cancer, prophylactic cranial irradiation, risk factors, brain metastases, overall survival

## Abstract

**Simple Summary:**

The brain is a common metastasis site of small-cell lung cancer (SCLC), and up to 50% of SCLC patients are at risk of brain metastasis (BM) within 2 years. Prophylactic cranial irradiation (PCI), as an essential treatment for reducing the risk of BM, inevitably leads to neurotoxicity. Differentiating the risk of BM in patients and individualized PCI treatment decisions may play an essential role in reducing the occurrence of BM, prolonging the overall survival, and improving the quality of life. Our study constructed and validated a clinical model to predict the incidence of BM and risk stratification for individualized PCI decisions. PCI could reduce the incidence of BM and improve overall survival (OS) in patients with a high risk of BM, but there was no significant difference between PCI and non-PCI groups in patients within a low-risk cohort.

**Abstract:**

Prophylactic cranial irradiation (PCI), as an essential part of the treatment of limited-stage small-cell lung cancer (LS-SCLC), inevitably leads to neurotoxicity. This study aimed to construct a brain metastasis prediction model and identify low-risk patients to avoid PCI; 236 patients with LS-SCLC were retrospectively analyzed and divided into PCI (63 cases) and non-PCI groups (173 cases). The nomogram was developed based on variables determined by univariate and multivariate analyses in the non-PCI group. According to the cutoff nomogram score, all patients were divided into high- and low-risk cohorts. A log-rank test was used to compare the incidence of brain metastasis between patients with and without PCI in the low-risk and high-risk groups, respectively. The nomogram included five variables: chemotherapy cycles (ChT cycles), time to radiotherapy (RT), lactate dehydrogenase (LDH), pro-gastrin-releasing peptide precursor (ProGRP), and lymphocytes–monocytes ratio (LMR). The area under the receiver operating characteristics (AUC) of the nomogram was 0.763 and 0.782 at 1 year, and 0.759 and 0.732 at 2 years in the training and validation cohorts, respectively. Based on the nomogram, patients were divided into high- and low-risk groups with a cutoff value of 165. In the high-risk cohort, the incidence of brain metastasis in the non-PCI group was significantly higher than in the PCI group (*p* < 0.001), but there was no difference in the low-risk cohort (*p* = 0.160). Propensity score-matching (PSM) analysis showed similar results; the proposed nomogram showed reliable performance in assessing the individualized brain metastasis risk and has the potential to become a clinical tool to individualize PCI treatment for LS-SCLC.

## 1. Introduction

Lung cancer has malignant tumors with the highest mortality, of which small-cell lung cancer (SCLC) accounts for approximately 15% [1,2]. SCLC is characterized by a short doubling time and early development of widespread metastases. The brain is a common metastasis site of SCLC, and up to 50% of SCLC patients are at risk of brain metastases (BM) within two years [3]. Prophylactic cranial irradiation (PCI) can reduce the incidence of BM and improve overall survival (OS) in patients with limited-stage SCLC (LS-SCLC) [4,5,6,7,8,9].

However, the neurocognitive dysfunction caused by PCI is a clinical issue that cannot be ignored [10]. Researchers proposed hippocampal avoidance PCI (HA-PCI) to reduce the adverse events of PCI [11,12,13]. However, recent studies showed that HA-PCI could not avoid declining neurocognitive functions [14,15]. Some researchers also suggested that brain magnetic resonance imaging follow-up combined with salvage stereotactic radiotherapy can replace PCI. However, high-quality evidence still fails to confirm its clinical feasibility [16,17,18,19]. In addition to the imprecision of their evidence, the above two methods also ignore the individualized differences between patients.

Brain metastasis does not occur in all patients with LS-SCLC. If we find a prediction model to assess BM risk accurately, patients with a low risk of BM will not receive PCI, thus avoiding neurotoxicity. Chung et al. [20] found, through risk stratification based on the presence of extra-thoracic metastases, hypermetabolism of bone marrow or spleen on FGD PET, and high neutrophil-to-lymphocyte ratio, that PCI could significantly benefit patients with an increased risk of BM, but not patients with low risk of BM. This research provides theoretical guidance for the clinical individualization of PCI in extensive-stage SCLC (ES-SCLC). However, how to differentiate patients’ risk of developing BM and individualize PCI treatment in LS-SCLC remains unknown. This study aimed to establish a nomogram model of BM prediction in LS-SCLC and optimize PCI treatment by stratifying the risk factor.

## 2. Patients and Method

### 2.1. Patients

In this retrospective study, data from patients with LS-SCLC confirmed by imaging and pathology from January 2012 to December 2018 in the Shanxi Provincial Cancer Hospital were collected. Medical records of baseline clinical characteristics were retrieved and reviewed from the Electronic Medical Record System, including chest and abdominal computed tomography (CT), neck and abdominal ultrasound, brain MRI, laboratory data, and prognostic-related factors identified in previous studies [21,22,23,24,25,26,27,28,29,30,31]. The clinical stage was classified using the tumor, node, metastases (TNM) system proposed by the American Joint Committee on Cancer (AJCC, the 8th edition). After treatment, patients were routinely followed up every 3–6 months by brain MRI.

The inclusion criteria were as follows: (1) complete clinical, laboratory, and follow-up data; (2) received two cycles of platinum-based, standard first-line chemotherapy. The exclusion criteria were: (1) previous anti-tumor therapy; (2) brain MRI was not performed during treatment and follow-up; (3) the follow-up time was less than 12 months; (4) history of another primary malignancy. Two hundred and forty-six patients met this study’s inclusion and exclusion criteria, of which 78 received PCI and 168 did not. The non-PCI group was randomly divided into training (*n* = 121) and validation cohorts (*n* = 47) for the establishment of the BM prediction model (Figure 1).

### 2.2. Nomogram Development

The differences between the stratification factors in the training cohort were analyzed using the log-rank test. We included the factors with *p* < 0.10 in the multivariate Cox regression analysis. The final model was formulated based on multivariate Cox regression analysis results using backward stepwise elimination with Akaike information criteria (AIC) as the stopping rule. Cox regression coefficients were utilized to generate a nomogram for predicting the incidence of BM. Finally, we developed a dynamic program from the normal nomogram to be better used in clinics.

### 2.3. Nomogram Model Validation and Risk Stratification of BM

We quantified the discriminative capability of the nomogram model by calculating the area under the receiver operating characteristics curve (ROC-AUC) with a 95% confidence interval (95% CI). The calibration curve was plotted using 1000 bootstrap resamples to evaluate the consistency between the predicted and actual probability of BM. Decision curve analysis (DCA) was used to assess the clinical benefit of the nomogram model by quantifying the net benefits at different threshold probabilities. High- and low-risk cohorts were stratified by the cutoff value of the total scores.

### 2.4. Statistical Analysis

In this study, TTBM was defined as the time from the first brain MRI to the occurrence of BM or the last MRI. Overall survival (OS) was calculated from the initial diagnosis to death of any cause or the last follow-up. We dichotomized continuous variables using the maximally selected rank statistics according to the optimal cutoff values. We balanced the potential confounding factors between comparable groups by the propensity score-matching (PSM) method. Statistical analyses were performed using R software (version 3.6.3; https://www.r-project.org/; accessed on 29 February 2020).

## 3. Results

### 3.1. Patient Characteristics

The clinical characteristics of the 246 patients who met the criteria in this study are summarized in Table 1. A total of 78 patients (31.7%) received PCI, while 168 (68.3%) patients did not. The median follow-up time was 26.5 months (5.0–115.8 months). A total of 104 (42.3%) patients developed BM during the follow-up period, and the median TTBM was 37.2 months (95% CI: 23.2–51.1 months). The median OS of the whole, non-PCI, and PCI groups was 43.3, 36.5, and 50.3 months, respectively. The incidence of BM in each group was 22.3%, 31.7%, and 0% at 1 year and 41.6%, 52.6%, and 14.3% at 2 years, respectively. We randomly divided non-PCI patients into training and validation cohorts, and there was no statistically significant difference for each factor (Appendix A).

### 3.2. Factors Predictive of BM

Several factors were significantly associated with brain metastases in patients with LS-SCLC (Appendix A). Multivariable Cox analysis demonstrated that clinical stage (III vs. I–II, HR = 3.91, 95% CI: 1.20–12.76, *p* = 0.024), ChT cycles (≥4 vs. <4, HR = 0.39, 95% CI: 0.20–0.75, *p* = 0.005), time to radiotherapy (Time to RT) (chemotherapy vs. Time to RT ≥ 1.8, HR: 0.66, 95% CI: 0.39–1.11, Time to RT < 1.8 vs. Time to RT ≥ 1.8, HR: 0.24, 95% CI: 0.08–0.69, *p* = 0.016), lactate dehydrogenase (LDH) (high vs. low, HR = 2.09, 95% CI: 1.24–3.53, *p* = 0.006), pro-gastrin-releasing peptide precursor (ProGRP) (high vs. low, HR = 2.29, 95% CI: 1.12–4.68, *p* = 0.023), and lymphocytes–monocytes ratio (LMR) (high vs. low, HR = 0.46, 95% CI: 0.26–0.79, *p* = 0.005) were significantly independent prognostic factors for TTBM (Figure 2).

### 3.3. Construction and Validation of a Nomogram

A nomogram was constructed to predict the incidence of BM based on identified prognostic factors (Stage III = 96 point, CT cycle < 4 = 67 point, Time to RT ≥ 2.9 = 100 point, chemotherapy = 70 point, high LDH = 52 point, high ProGRP = 58 point, low LMR = 55 point, and others = 0 point) (Figure 3). According to the contribution degree of each factor in the nomogram, an individual’s total score is calculated. The calibration curve of the nomogram for the probability of one- and two-year BM demonstrated a good consistency between the nomogram and actual observation in both the training and validation cohorts (Figure 4). The AUC-ROC of the nomogram predicted that the incidence of BM in the training and validation cohort was 0.782 (95% CI: 0.696–0.868) and 0.731 (95% CI: 0.584–0.878) at 1 year, and 0.725 (95% CI: 0.619–0.832) and 0.732 (95% CI: 0.582–0.731) at 2 years, respectively (Figure 5). The DCA shows that if the threshold probability were between 3.4% and 89.0% with 1 year, and between 7.8% and 73.6% with 2 years, then using the nomogram to predict the brain metastasis probability in LS-SCLC patients would be beneficial (Figure 6).

### 3.4. Risk Stratification of BM

All patients were divided into low- and high-risk cohorts based on the nomogram score at the cutoff value of 177. The incidence of BM in the low- and high-risk cohorts had a statistically significant difference (*p* < 0.001). In the high-risk cohort, PCI can not only significantly decrease the incidence of BM (*p* < 0.001) but can also significantly increase the OS (*p* = 0.016) (Figure 7 and Figure 8). However, PCI could not substantially reduce the BM incidence (*p* = 0.630) and improve OS (*p* = 0.690) in the low-risk cohort (Figure 7 and Figure 8). After being matched by PSM, PCI can still decrease the incidence of BM (*p* < 0.001) and increase the OS rate (*p* = 0.022) in the high-risk cohort. Moreover, there was no statistically significant difference in the incidence of BM (*p* = 0.620) and OS rate (*p* = 0.670) in the low-risk cohort (Figure 7 and Figure 8).

## 4. Discussion

PCI can significantly reduce the incidence of BM and improve OS in patients with limited-stage SCLC. However, the decline in neurocognitive function reduces health-related quality of life. Individualized PCI therapy according to the risk of BM can benefit more LS-SCLC patients. This study demonstrated that clinical stage, ChT cycles, Time to RT, LDH, ProGRP, and LMR were the independent influencing factors of TTBM. By combining the selected risk factors, we constructed and validated a reliable tool to predict the incidence of BM. The calibration curve indicated an excellent consistency between predictions and observations for the nomogram. The cumulative incidence of BM was significantly different between the low- and high-risk cohorts in the non-PCI group. PCI could substantially reduce the incidence of BM and improve OS in patients with a high risk of BM, but there was no significant difference between PCI and non-PCI groups in patients within the low-risk cohort.

Although previous studies have explored the risk factors of BM in SCLC, there are some deficiencies [20,32,33,34,35,36,37,38,39,40,41]. First, brain MRI was not used as a standard measure and was only performed with symptoms. For this reason, some patients with brain metastasis were included in these studies, resulting in the inability to accurately evaluate the related factors of brain metastasis [32,35,36,37,38,39,40]. Second, some studies have taken PCI as a factor related to brain metastasis. As PCI can significantly reduce the risk of brain metastasis, some factors related to BM may be ignored due to the inclusion of PCI [41]. Third, although some risk factors have been found to predict brain metastasis in patients with SCLC, no risk stratification has been developed to optimize PCI selection in LS-SCLC patients [32,35,36,37]. Finally, some previous studies focused on brain metastases in ES-SCLC patients rather than LS-SCLC [20,33,34]. To ensure the accuracy and application value of the nomogram, we constructed and verified the nomogram in the non-PCI cohort in which all LS-SCLC patients underwent brain MRIs.

The clinical stage was significantly associated with the prognosis for patients with LS-SCLC. Gong et al. [32] also found an increased frequency of BM in stage III compared to stage I–II in LS-SCLC (HR: 2.458, *p* = 0.002). Wu et al. [37] found that the cumulative incidence of BM for stages III and I–II was 21% and 10% at two years, respectively. The multivariate analyses showed that the stage was the only factor independently associated with BM (HR: 2.09; 95% CI: 1.08–4.04; *p* = 0.028). Our study also found that the incidence of BM in LS-SCLC patients with stage III was 3.91 times that of patients with stage I–II. To our knowledge, chemo-radiotherapy is the standard treatment for patients with LS-SCLC. However, whether ChT cycles and radiotherapy are related to BM remains controversial [35,36]. Less than four chemotherapy cycles were an independent risk factor for BM (HR = 0.49, *p* = 0.036), which may be due to the failure to kill micro-metastases with fewer ChT cycles [36]. Our study also demonstrated that the number of ChT cycles was significantly associated with BM. In addition, this study also found that patients with a shorter interval between chemotherapy and radiotherapy had a lower risk of BM, which is consistent with previous conclusions [42,43,44]. The NCCN guidelines strongly recommend starting radiotherapy as soon as possible during the first to second chemotherapy cycle [8]. Our study also found that the patients who received subsequent thoracic radiotherapy had a higher risk of BM than those who only received chemotherapy (Figure 2). The most likely reason was that some physicians were unfamiliar with the radiotherapy indication. In their opinion, for patients who achieved a good response after chemotherapy, radiotherapy was unnecessary.

We also found that lower LMR, higher LHD, and ProGRP were independent risk factors for BM in LS-SCLC patients. Numerous studies found that LDH and ProGRP levels reflected systemic tumor burden, which might be associated with the risk of treatment failure, including brain metastases [27,28,45]. ProGRP, a neuropeptide that might be involved in several physiological functions, was reported as a high-sensitivity and high-specificity marker for diagnosis, treatment monitoring, and survival in SCLC. Yonemori et al. [45] found that ProGRP was a significant predictive factor for the first failure event due to BM (HR: 12.5, 95% CI: 2.00–77.9, *p* = 0.007) and the overall incidence of BM (HR: 5.89, 95% CI: 1.25–27.7, *p* = 0.025) in LS-SCLC. Kazuihto et al. [21] also demonstrated that pretreatment ProGRP levels could help predict the development of BM in LS-SCLC patients and identify which patients benefit from PCI. Though there was no report on the association between LDH and brain metastasis in SCLC, some researchers found that elevated LDH was frequently associated with poor survival outcomes in SCLC [46,47,48]. The mechanism is unclear, and it is assumed that LDH promotes tumor metastases by participating in tumor angiogenesis and immune escape [27,49]. Decreased levels of LMR, a marker of systemic inflammation, could be used as a poor prognostic index in some types of cancers [50,51,52]. Our work confirmed that patients with low LMR were prone to brain metastases, consistent with previous studies.

Although PCI is accepted as a standard treatment for LS-SCLC, some studies have found that PCI is not associated with OS benefits [16,18]. This contradictory result may be closely related to whether the brain MRI screening is performed before PCI and the different probabilities of BM in various subjects. Our study established a BM prediction model and divided the patients into high-risk and low-risk cohorts. PCI significantly reduced the risk of BM and improved OS for patients in the high-risk cohort, but no clear benefit of PCI was found in the low-risk cohort. Additionally, MRI surveillance combined with post-BM SRS can achieve a prognosis similar to PCI, but the evidence is insufficient. This study provided a stratification based on the nomogram, which showed PCI was beneficial only for patients with a high risk of BM. It may have the potential to help determine the appropriate treatment strategy for LS-SCLC patients.

There are still some limitations to this study. The most significant being that it is a small retrospective study design, not a randomized controlled trial, limited by inherent defects. Secondly, the number of patients enrolled in this study is small due to it being limited to a single center and strict inclusion criteria. This study aimed to investigate the factors affecting BM. Therefore, excluding patients who died before BM with 12 months or less than 12 months of follow-up time may have inferior performance in identifying patients at risk for early death. Further large-scale prospective studies are needed to confirm the validity of our results.

## 5. Conclusions

Our study showed that clinical stage, CT cycle, Time to RT, LDH, ProGRP, and LMR were associated with BM incidence in LS-SCLC. The nomogram-based model has the potential to help clinicians evaluate the risk stratification of BM and avoid performing unnecessary PCI.

## Figures and Tables

**Figure 1 cancers-14-04906-f001:**
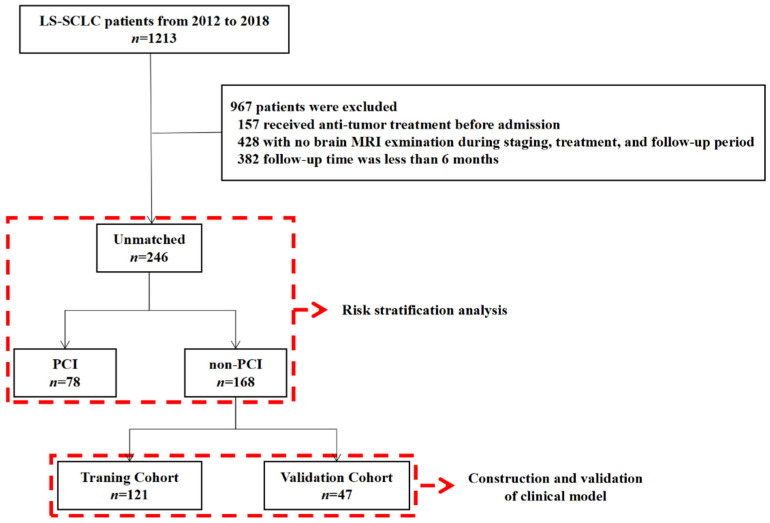
Diagram of patient enrollment and subgroup analysis. A total of 1213 patients with LS-SCLC met the inclusion criteria. Finally, only 246 patients were eligible to enroll in this study.

**Figure 2 cancers-14-04906-f002:**
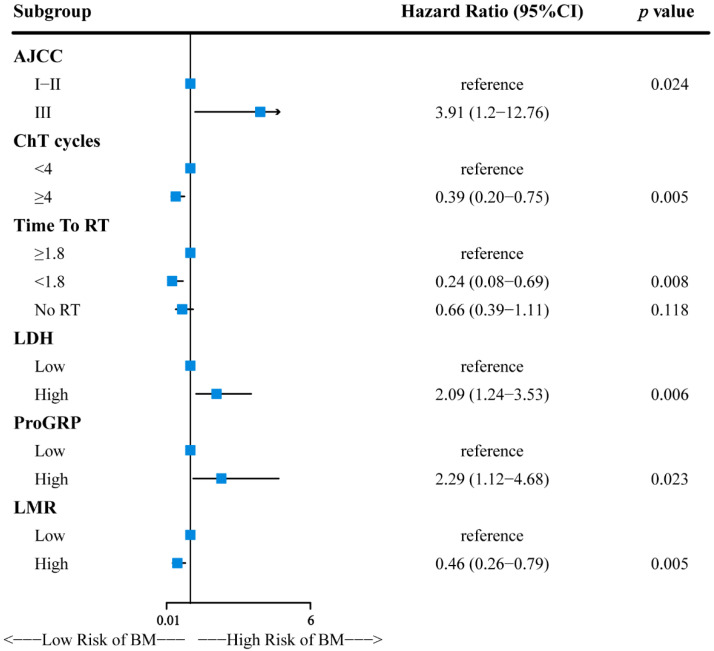
Forest plot for multivariate Cox regression analysis and 95% confidence intervals.

**Figure 3 cancers-14-04906-f003:**
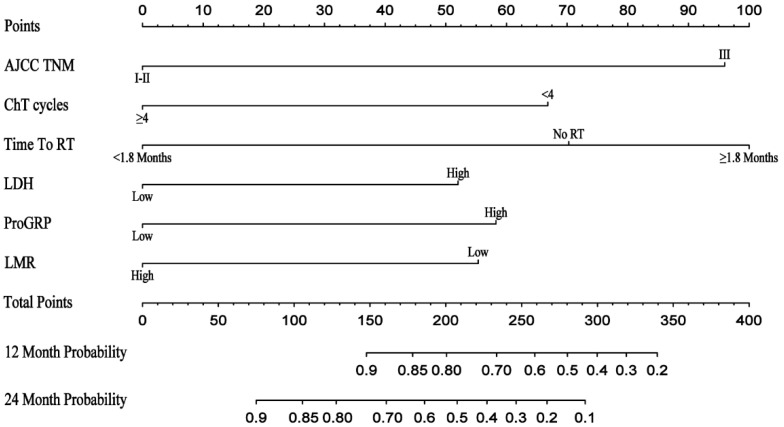
The nomogram was developed in the training cohort. ChT cycles, chemotherapy cycles; Time to RT, time from chemotherapy to radiotherapy; LDH, lactate dehydrogenase; ProGRP, pro-gastrin-releasing peptide precursor; LMR, lymphocytes–monocytes ratio.

**Figure 4 cancers-14-04906-f004:**
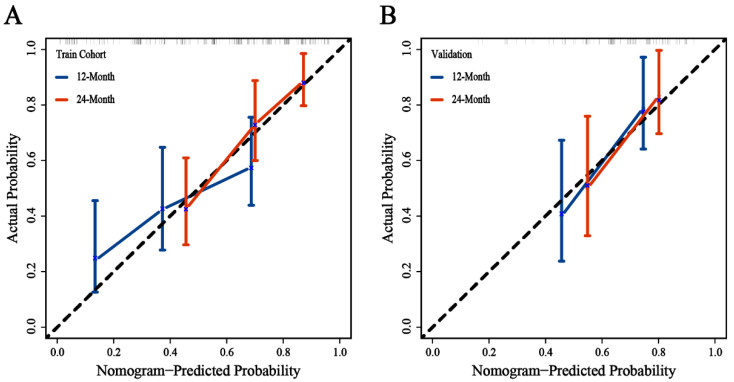
Calibration curves of the nomogram prediction in training (**A**) and validation cohort (**B**).

**Figure 5 cancers-14-04906-f005:**
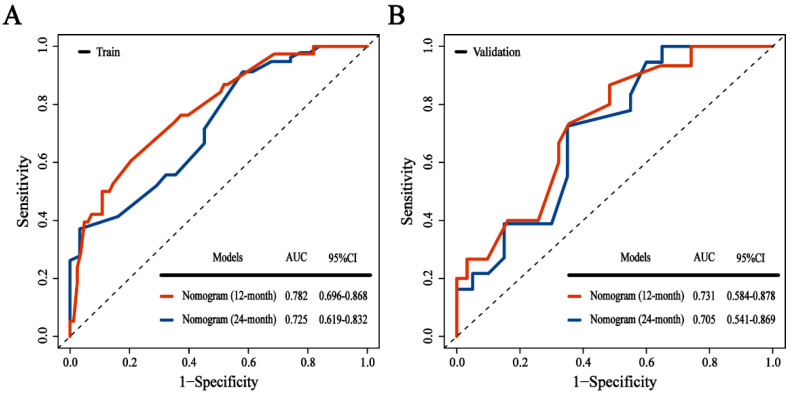
Performance comparison of the Nomogram Score. ROC curve in training (**A**) and validation cohort (**B**). ROC, receiver operating characteristics; AUC, area under the ROC; CI, confidence interval.

**Figure 6 cancers-14-04906-f006:**
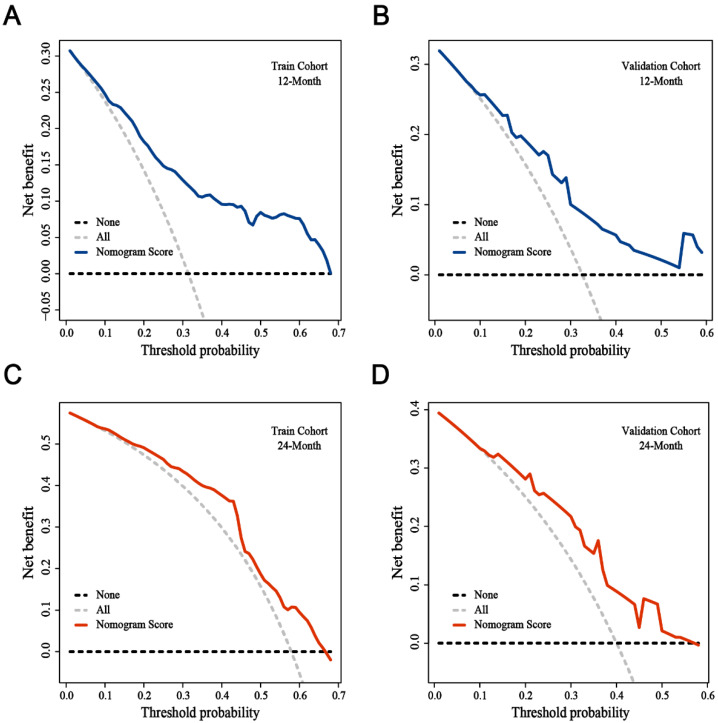
Decision curve analysis for the nomogram in training (**A**,**C**) and validation cohort (**B**,**D**).

**Figure 7 cancers-14-04906-f007:**
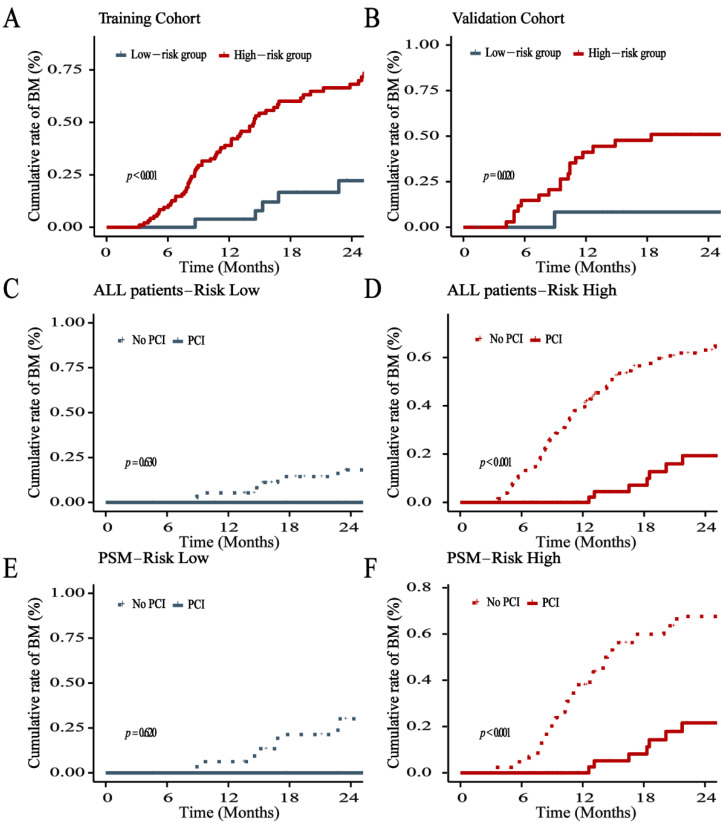
Proportion of patients with brain metastases in training (**A**) and validation (**B**), low-risk cohort (**C**), and high-risk cohort (**D**) before matched. The proportion of patients with brain metastasis in the low-risk and high-risk cohorts after PSM (**E**,**F**).

**Figure 8 cancers-14-04906-f008:**
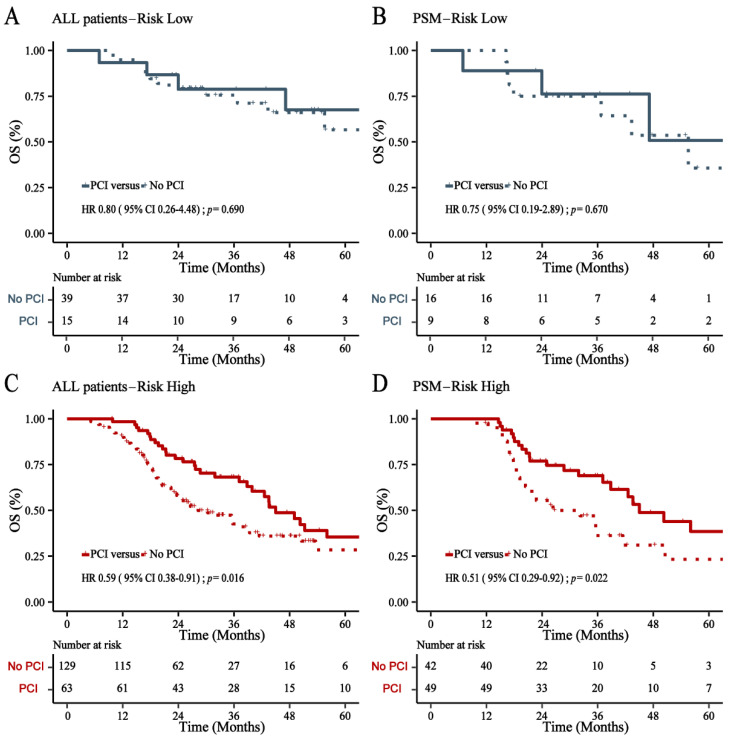
Overall survival (OS) in the low-risk cohort (**A**) and high-risk cohort (**C**) before matched, and after PSM (**B**,**D**).

**Table 1 cancers-14-04906-t001:** Patient characteristics before and after PSM.

Variable	Before PSM				After PSM			
	All Patients	Non-PCI	PCI	*p*	All Patients	Non-PCI	PCI	*p*
Clinical stage				0.192				0.186
I–II	21 (8.5%)	17 (10.12%)	4 (5.13%)		10 (8.6%)	7 (12.07%)	3 (5.17%)	
III	225 (91.5%)	151 (89.88%)	74 (94.87%)		106 (91.4%)	51 (87.93%)	55 (94.83%)	
Gender				0.532				0.83
Male	57 (23.2%)	37 (22.02%)	20 (25.64%)		29 (25%)	15 (25.86%)	14 (24.14%)	
Female	189 (76.8%)	131 (77.98%)	58 (74.36%)		87 (75%)	43 (74.14%)	44 (75.86%)	
Age				0.052				0.353
<70	129 (52.4%)	81 (48.21%)	48 (61.54%)		61 (52.6%)	28 (48.28%)	33 (56.9%)	
≥70	117 (47.6%)	87 (51.79%)	30 (38.46%)		55 (47.4%)	30 (51.72%)	25 (43.1%)	
ECOG				0.036				0.717
0–1	214 (87%)	141 (83.93%)	73 (93.59%)		108 (93.1%)	55 (94.83%)	53 (91.38%)	
2–4	32 (13%)	27 (16.07%)	5 (6.41%)		8 (6.9%)	3 (5.17%)	5 (8.62%)	
Smoke				0.175				0.555
Yes	74 (30.1%)	46 (27.38%)	28 (35.9%)		39 (33.6%)	18 (31.03%)	21 (36.21%)	
No	172 (69.9%)	122 (72.62%)	50 (64.1%)		77 (66.4%)	40 (68.97%)	37 (63.79%)	
BMI				0.099				0.85
Normal	154 (62.6%)	111 (66.07%)	43 (55.13%)		69 (59.5%)	35 (60.34%)	34 (58.62%)	
Abnormal	92 (37.4%)	57 (33.93%)	35 (44.87%)		47 (40.5%)	23 (39.66%)	24 (41.38%)	
ChT cycles				0.033				1.000
<4	24 (9.8%)	21 (12.5%)	3 (3.85%)		3 (2.6%)	2 (3.45%)	1 (1.72%)	
≥4	222 (90.2%)	147 (87.5%)	75 (96.15%)		113 (97.4%)	56 (96.55%)	57 (98.28%)	
Time To RT				<0.001				0.775
≥1.8	157 (72.4%)	96 (57.1%)	61 (78.2%)		91 (78.5%)	44 (75.86%)	47 (81.03%)	
<1.8	60 (24.4%)	55 (32.7%)	5 (6.4%)		12 (10.3%)	7 (12.07%)	5 (8.62%)	
No RT	29 (11.8%)	17 (10.1%)	12 (15.4%)		12 (11.2%)	7 (12.07%)	6 (10.34%)	
HGB				0.024				1.000
Normal	16 (6.5%)	15 (8.93%)	1 (1.28%)		3 (2.6%)	2 (3.45%)	1 (1.72%)	
Abnormal	230 (93.5%)	153 (91.07%)	77 (98.72%)		113 (97.4%)	56 (96.55%)	57 (98.28%)	
Na				<0.001				0.834
Normal	182 (74%)	136 (80.95%)	46 (58.97%)		85 (73.3%)	42 (72.41%)	43 (74.14%)	
Abnormal	64 (26%)	32 (19.05%)	32 (41.03%)		31 (26.7%)	16 (27.59%)	15 (25.86%)	
LYM				0.713				0.802
Low	205 (83.3%)	139 (82.74%)	66 (84.62%)		97 (83.6%)	49 (84.48%)	48 (82.76%)	
High	41 (16.7%)	29 (17.26%)	12 (15.38%)		19 (16.4%)	9 (15.52%)	10 (17.24%)	
PLT				0.23				0.793
Low	46 (18.7%)	28 (16.67%)	18 (23.08%)		17 (14.7%)	8 (13.79%)	9 (15.52%)	
High	200 (81.3%)	140 (83.33%)	60 (76.92%)		99 (85.3%)	50 (86.21%)	49 (84.48%)	
MPV				0.52				0.678
Low	73 (29.7%)	52 (30.95%)	21 (26.92%)		32 (27.6%)	17 (29.31%)	15 (25.86%)	
High	173 (70.3%)	116 (69.05%)	57 (73.08%)		84 (72.4%)	41 (70.69%)	43 (74.14%)	
LDH				0.363				0.455
Low	122 (49.6%)	80 (47.62%)	42 (53.85%)		64 (55.2%)	34 (58.62%)	30 (51.72%)	
High	124 (50.4%)	88 (52.38%)	36 (46.15%)		52 (44.8%)	24 (41.38%)	28 (48.28%)	
AGR				0.209				0.059
Low	177 (72%)	125 (74.4%)	52 (66.67%)		85 (73.3%)	47 (81.03%)	38 (65.52%)	
High	69 (28%)	43 (25.6%)	26 (33.33%)		31 (26.7%)	11 (18.97%)	20 (34.48%)	
UA				0.944				0.166
Low	31 (12.6%)	21 (12.5%)	10 (12.82%)		15 (12.9%)	10 (17.24%)	5 (8.62%)	
High	215 (87.4%)	147 (87.5%)	68 (87.18%)		101 (87.1%)	48 (82.76%)	53 (91.38%)	
CysC				0.626				0.555
Low	83 (33.7%)	55 (32.74%)	28 (35.9%)		39 (33.6%)	21 (36.21%)	18 (31.03%)	
High	163 (66.3%)	113 (67.26%)	50 (64.1%)		77 (66.4%)	37 (63.79%)	40 (68.97%)	
CEA				0.007				0.529
Low	153 (62.2%)	95 (56.55%)	58 (74.36%)		85 (73.3%)	44 (75.86%)	41 (70.69%)	
High	93 (37.8%)	73 (43.45%)	20 (25.64%)		31 (26.7%)	14 (24.14%)	17 (29.31%)	
NSE				0.046				1.000
Low	219 (89%)	145(86.31%)	74 (94.87%)		108 (93.1%)	54 (93.1%)	54 (93.1%)	
High	27 (11%)	23(13.69%)	4 (5.13%)		8 (6.9%)	4 (6.9%)	4 (6.9%)	
ProGRP				0.732				1.000
Low	217 (88.2%)	149 (88.69%)	68 (87.18%)		100 (86.2%)	50 (86.21%)	50 (86.21%)	
High	29 (11.8%)	19 (11.31%)	10 (12.82%)		16 (13.8%)	8 (13.79%)	8 (13.79%)	
CA125				0.13				0.075
Low	201 (81.7%)	133 (79.17%)	68 (87.18%)		90 (77.6%)	41 (70.69%)	49 (84.48%)	
High	45 (18.3%)	35 (20.83%)	10 (12.82%)		26 (22.4%)	17 (29.31%)	9 (15.52%)	
NLR				0.45				0.431
Low	31 (12.6%)	23 (13.69%)	8 (10.26%)		17 (14.7%)	10 (17.24%)	7 (12.07%)	
High	215 (87.4%)	145 (86.31%)	70 (89.74%)		99 (85.3%)	48 (82.76%)	51 (87.93%)	
PLR				0.583				0.542
Low	210 (85.4%)	142 (84.52%)	68 (87.18%)		104 (89.7%)	51 (87.93%)	53 (91.38%)	
High	36 (14.6%)	26 (15.48%)	10 (12.82%)		12 (10.3%)	7 (12.07%)	5 (8.62%)	
ALI				0.126				0.636
Low	211 (85.8%)	148 (88.1%)	63 (80.77%)		94 (81%)	48 (82.76%)	46 (79.31%)	
High	35 (14.2%)	20 (11.9%)	15 (19.23%)		22 (19%)	10 (17.24%)	12 (20.69%)	
SIRI				0.009				1.000
Low	213 (86.6%)	139 (82.74%)	74 (94.87%)		109 (94%)	54 (93.1%)	55 (94.83%)	
High	33 (13.4%)	29 (17.26%)	4 (5.13%)		7 (6%)	4 (6.9%)	3 (5.17%)	
AAPR				0.913				1.000
Low	220 (89.4%)	150 (89.29%)	70 (89.74%)		106 (91.4%)	53 (91.38%)	53 (91.38%)	
High	26 (10.6%)	18 (10.71%)	8 (10.26%)		10 (8.6%)	5 (8.62%)	5 (8.62%)	
PNI				0.046				0.342
Low	31 (12.6%)	26 (15.48%)	5 (6.41%)		11 (9.5%)	7 (12.07%)	4 (6.9%)	
High	215 (87.4%)	142 (84.52%)	73 (93.59%)		105 (90.5%)	51 (87.93%)	54 (93.1%)	
LMR				0.172				0.608
Low	54 (22%)	41 (24.4%)	13 (16.67%)		18 (15.5%)	10 (17.24%)	8 (13.79%)	
High	192 (78%)	127 (75.6%)	65 (83.33%)		98 (84.5%)	48 (82.76%)	50 (86.21%)	

ECOG, Eastern Cooperative Oncology Group; ChT cycles, chemotherapy cycles; Time To RT, time from chemotherapy to radiotherapy; BMI, body mass index; HGB, hemoglobin; LYM, lymphocyte; PLT, platelet; MPV, men platelet volume; LDH, lactate dehydrogenase; AGR, albumin-to-globulin ratio; UA, uric acid; CysC, cystatin C; CEA, carcinoembryonic antigen; NSE, neuron-specific enolase; ProGRP, pro-gastrin-releasing peptide precursor; CA125, carbohydrate antigen 125; NLR, neutrophil-to-lymphocyte ratio; PLR, platelet-to-lymphocyte ratio; ALI, advanced lung cancer inflammation index; SIRI, systemic inflammation response index; AAPR, albumin-to-alkaline phosphatase ratio; PNI, prognostic nutrition index; LMR, lymphocytes–monocytes ratio.

## Data Availability

The data generated in this study are not publicly available for ethical reasons. Data access is, however, possible upon reasonable request to the corresponding authors.

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
