# Peer review of "Construction of Brain Metastasis Prediction Model and Optimization of Prophylactic Cranial Irradiation Selection for Limited-Stage Small-Cell Lung Cancer"

_cancers, 2022, doi:10.3390/cancers14194906_

Round 1

Reviewer 1 Report

This is an interesting, mono-center retrospective study evaluating the clinical stage, radiologic reports (brain MR, etc), and laboratory data of patients with small cell lung carcinoma (SCLC) to establish a nomogram model of BM prediction in the limited stage of SCLC end optimize PCI treatment.

 However, some changes are needed to improve the study.

 Minor comments:

1.)    Page 1, Line 10 In the simple summary: The meaning of the abbreviation SCLC should be explained and put in brackets

2.)    Page 1, Line 16 In the simple summary: The meaning of the abbreviation OS should be explained and put in brackets

3.)    Page 1 Abstract should be without headings according to guidelines

4.) Abstract should be without references (lines 21 and 27)

5.) Page 1, Line 28: various abbreviations are used for the term chemotherapy cycles- ChT or CT

6.) Page 3:In patients: time interval is from 2012. to 2019, and in Figure 1 – diagram: it stays from 2013. until 2018, – Which is correct?

7.) Page 4:In table 1: The meaning of the abbreviation ECOG should be explained below the table

8.) Page 11:Abstract - The meaning of the abbreviation ES – SCLC

9.) References should be checked: for example reference 7 (line 334), there are two brackets without a number

Major comments:

1) The title states the phrase: under MRI monitoring - stage I. Nowhere in the text is MR monitoring (time interval or similar) particularly emphasized as an important variable. Stage I and II together are also used as a clinical indicator, while it is not highlighted anywhere. I think both should be left out of the title.

2) There are several studies on small lung carcinoma that have made a nomogram related to survival and cancer stage, which are not cited. In this paper, the advantage of this nomogram for patients is not sufficiently highlighted, i.e. whether it is exclusively for the maintenance of cognitive abilities or the overall survival of patients, and what the nomogram specifically means to clinicians. That is, it is not clarified what the scientific contribution of this study is compared to previous ones that also developed similar nomograms related to the topic of small cell lung carcinoma, brain metastasis, and overall survival. Also, those studies were not cited and should be mentioned in the discussion.

For example:

Shan Q, Shi J, Wang X, Guo J, Han X, Wang Z, Wang H. A new nomogram and risk classification system for predicting survival in small cell lung cancer patients diagnosed with brain metastasis: a large population-based study. BMC Cancer. 2021 May 29;21(1):640. doi: 10.1186/s12885-021-08384-5. PMID: 34051733; PMCID: PMC8164795.

Jiang A, Liu N, Zhao R, Liu S, Gao H, Wang J, Zheng X, Ren M, Fu X, Liang X, Tian T, Ruan Z, Yao Y. Construction and Validation of a Novel Nomogram to Predict the Overall Survival of Patients With Combined Small Cell Lung Cancer: A Surveillance, Epidemiology, and End Results Population-Based Study. Cancer Control. 2021 Jan-Dec;28:10732748211051228. doi: 10.1177/10732748211051228. PMID: 34632799; PMCID: PMC8512214.

Yang Y, Sun S, Wang Y, Xiong F, Xiao Y, Huang J. Development and validation of nomograms for predicting survival of elderly patients with stage I small-cell lung cancer. Bosn J Basic Med Sci. 2021 Oct 1;21(5):632-641. doi: 10.17305/bjbms.2020.5420. PMID: 33577444; PMCID: PMC8381200.

3) Did this study analyze PHD because there is tumor variability there, which could affect the occurrence of brain metastases and patient survival?

Author Response

Response to Reviewer 1 Comments

Thank you for offering us an opportunity to improve the quality of our manuscript (1909661). We appreciated your constructive and insightful comments very much. In this revision, we have addressed all of these comments/suggestions. Any modifications to the manuscript are marked up using the “Track Changes” function.

Minor comments:

Point 1: Page 1, Line 10 In the simple summary: The meaning of the abbreviation SCLC should be explained and put in brackets

Response 1: The meaning of the abbreviation SCLC has been supplemented in the manuscript (page 1, line 10).

Point 2: Page 1, Line 16 In the simple summary: The meaning of the abbreviation OS should be explained and put in brackets

Response 2: The meaning of the abbreviation OS has been supplemented in the manuscript (page 1, line 17).

Point 3: Page 1 Abstract should be without headings according to guidelines

Response 3: Headings in abstract have been removed according to guidelines (page 1, line 19-37).

Point 4: Abstract should be without references (lines 21 and 27)

Response 4: References in abstract have been removed.

Point 5: Page 1, Line 28: various abbreviations are used for the term chemotherapy cycles- ChT or CT

Response 5: The full text and figures of the manuscript has been revised to cycles-ChT based on your prompts.

Point 6: Page 3: In patients: time interval is from 2012. to 2019, and in Figure 1 – diagram: it stays from 2013. until 2018, – Which is correct?

Response: 6 Thank you for your careful review of the manuscript. After careful verification of the data, the time interval for patient data collection was from 2012 to 2018 (page 2, line 77, page 3, Figure 1).

Point 7: Page 4: In table 1: The meaning of the abbreviation ECOG should be explained below the table

Response 7: The meaning of the abbreviation ECOG has been supplemented below the table (page 5, line 135).

Point 8: Page 11: Abstract - The meaning of the abbreviation ES – SCLC

Response 8: The meaning of the abbreviation ES-SCLC is on Page 2, Line 68.

Point 9: References should be checked: for example reference 7 (line 334), there are two brackets without a number

Response 9: The manuscript has been checked and revised for all references.

Major comments:

Point 1: The title states the phrase: under MRI monitoring - stage I. Nowhere in the text is MR monitoring (time interval or similar) particularly emphasized as an important variable. Stage I and II together are also used as a clinical indicator, while it is not highlighted anywhere. I think both should be left out of the title.

Response 1: Thanks for your suggestion, the title of the manuscript has been revised.

Point 2: There are several studies on small lung carcinoma that have made a nomogram related to survival and cancer stage, which are not cited. In this paper, the advantage of this nomogram for patients is not sufficiently highlighted, i.e. whether it is exclusively for the maintenance of cognitive abilities or the overall survival of patients, and what the nomogram specifically means to clinicians. That is, it is not clarified what the scientific contribution of this study is compared to previous ones that also developed similar nomograms related to the topic of small cell lung carcinoma, brain metastasis, and overall survival. Also, those studies were not cited and should be mentioned in the discussion.

For example:

Shan Q, Shi J, Wang X, Guo J, Han X, Wang Z, Wang H. A new nomogram and risk classification system for predicting survival in small cell lung cancer patients diagnosed with brain metastasis: a large population-based study. BMC Cancer. 2021 May 29;21(1):640. doi: 10.1186/s12885-021-08384-5. PMID: 34051733; PMCID: PMC8164795.

Jiang A, Liu N, Zhao R, Liu S, Gao H, Wang J, Zheng X, Ren M, Fu X, Liang X, Tian T, Ruan Z, Yao Y. Construction and Validation of a Novel Nomogram to Predict the Overall Survival of Patients With Combined Small Cell Lung Cancer: A Surveillance, Epidemiology, and End Results Population-Based Study. Cancer Control. 2021 Jan-Dec;28:10732748211051228. doi: 10.1177/10732748211051228. PMID: 34632799; PMCID: PMC8512214.

Yang Y, Sun S, Wang Y, Xiong F, Xiao Y, Huang J. Development and validation of nomograms for predicting survival of elderly patients with stage I small-cell lung cancer. Bosn J Basic Med Sci. 2021 Oct 1;21(5):632-641. doi: 10.17305/bjbms.2020.5420. PMID: 33577444; PMCID: PMC8381200.

Response 2: Several previous studies constructed nomograms based on conventional variables to predict the prognosis of small cell lung cancer. However, this study used more and more comprehensive variables to build nomograms to predict the occurrence of brain metastasis. Because the purpose of prediction is different, we do not cite the above literature.

Point 3: Did this study analyze PHD because there is tumor variability there, which could affect the occurrence of brain metastases and patient survival?

Response 3: Thank you very much for your suggestion. Does PHD mean prolyl-hydroxylase? If so, PHD is not a clinically measured indicator, so we cannot include PHD in the analysis.

Reviewer 2 Report

This issue is important, but there are some points to revise before publishing.

1 I think that main thema of this article is PCI for LD-SCLC. However, authors wrote about ED-SCLC in introduction. Why?

2 Authors said that prognostic factor idetified in previous studies in line75.Could you show referecies?

3 Could you show the reason of not-receving PCI?

4 Could you tell me the neurotoxicities differences between PCI and non-PCI group?

Author Response

Response to Reviewer 2 Comments

Thank you for offering us an opportunity to improve the quality of our manuscript (1909661). We appreciated your constructive and insightful comments very much. In this revision, we have addressed all of these comments/suggestions. Any modifications to the manuscript are marked up using the “Track Changes” function.

Point 1: I think that main thema of this article is PCI for LD-SCLC. However, authors wrote about ED-SCLC in introduction. Why?

Response 1: Thanks for your suggestion, we have rewritten the third paragraph of the introduction in the manuscript (page2, line 58-73).

Point 2: Authors said that prognostic factor identified in previous studies in line75.Could you show referecies?

Response 2: We have provided references for the prognostic factors identified in previous studies (page 2, line 81).

Point 3: Could you show the reason of not-receving PCI?

Response 3: We carefully reviewed the patient's medical records. There are two main reasons for not receiving PCI: first, patients refuse to accept PCI because of neurotoxicity; second, in this study, 11.8% of the patient’s received chemotherapy in the department of oncology but not radiotherapy. Due to the lack of understanding of radiotherapy by oncologists, prophylactic brain irradiation is not recommended to patients.

Point 4: Could you tell me the neurotoxicities differences between PCI and non-PCI group?

Response 4: Prophylactic cranial irradiation (PCI) can improve the prognosis of patients with limited small cell lung cancer and reduce the incidence of brain metastasis. Still, it inevitably leads to neurotoxicities, such as alopecia, headache, fatigue, nausea, memory loss, intellectual disability, dementia, and ataxia[1]. If patients do not receive PCI, there will be no neurotoxicity, but there is a high incidence of brain metastasis.

References

  1. Pechoux, C.L.; Sun, A.; Slotman, B.J.; De Ruysscher, D.; Belderbos, J.; Gore, E.M. Prophylactic cranial irradiation for patients with lung cancer. Lancet Oncol. 2016, 17, e277-e293. DOI:10.1016/S1470-2045(16)30065-1

Reviewer 3 Report

The authors proposed a retrospective study to evaluate a prediction model for patients with brain metastasis of Small-Cell Lung Cancer to identify a low-risk group to avoid Prophylactic Cranial Irradiation. The proposed nomogram showed a reliable performance in assessing the individualized brain metastasis risk and has the potential to become a clinical tool to individualize prophylactic treatment for low-risk SCLC. The manuscript is attractive and well conducted. The data results are an added value to clinical practice to stratify the patients into two classes of risk for the application of cranial prophylactic irradiation in patients with a high risk to develop brain metastases. The authors report the limitation of the study but the data and the nomogram in the clinical use could allow for stratifying the data of this patient setting and lead to a more extensive randomized study. I believe that the study can be accepted in the formula in which it is found. Thank you for allowing me to perform this interesting review.

Author Response

Thank you for your positive comments on our manuscript entitled “Construction of brain Metastasis Prediction Model and Optimization of Prophylactic cranial irradiation selection for limited-stage small Cell Lung Cancer under MRI Monitoring”. Your approval of this study is a great encouragement to us.

Round 2

Reviewer 1 Report

Explanation of point 3:

By the term PHD, I meant the pathohistological diagnosis, that is, the subclassification of tumors. The current sub-classification recognizes two subtypes: pure SCLC and combined SCLC. 

Reviewer 2 Report

I have no points to added comments.